

# pygeodyn 1.0.0: a Python package for geomagnetic data assimilation

Loïc Huder[1], Nicolas Gillet[1], and Franck Thollard[1]

[1]Univ. Grenoble Alpes, Univ. Savoie Mont Blanc, CNRS, IRD, IFSTTAR, ISTerre, 38000 Grenoble, France

**Correspondence:** Nicolas Gillet (nicolas.gillet@univ-grenoble-alpes.fr)

**Abstract.** `pygeodyn` is a sequential geomagnetic data assimilation package written in Python. It gives access to the core surface dynamics, controlled by geomagnetic observations, by means of a stochastic model anchored to geodynamo simulation statistics. `pygeodyn` aims at giving access to a user-friendly and flexible data assimilation algorithm. It is designed to be tunable by the community by different means: possibility to use embedded data and priors, or to supply custom ones; tunable

parameters through configuration files; adapted documentation for several user profiles. In addition, output files are directly supported by the package `webgeodyn` that provides a set of visualisation tools to explore the results of computations.

## 1 Introduction

The magnetic field of the Earth is generated by motions of liquid metal in the outer core, a process called the "geodynamo".

To tackle this complex problem, direct numerical simulations (DNS) have been developed to model the coupling between the primitive equations for heat, momentum and induction in a rotating spherical shell. With the development of computing power, DNS capture more and more of the physics thought to be at work in the Earth core (degree of dipolarity, ratio of magnetic to kinetic energy, occurrence of torsional Alfvén waves, etc. see for instance Schaeffer et al., 2017). However, despite such advances, the geodynamo problem is so challenging that DNS are not suited yet to reproduce changes observed at interannual

periods with modern data (e.g. Finlay et al., 2016). Simulating numerically dynamo action at Earth-like rotation rates indeed requires resolving time-scales $10^8$ orders of magnitude apart (from a fraction of day to 10 kyrs) for $N \approx (10^6)^3$ degrees of freedom. This requirement is unlikely to be satisfied in a nearby future with DNS, making the prediction of the geomagnetic field evolution an extremely challenging task. For these reasons, promising strategies involving large-eddy simulations (LES, see Aubert et al., 2017) are emerging, but these are currently unable to ingest recent geophysical records.

Many efforts were devoted to the improvement of observable geodynamo quantities: the magnetic field above the surface of the Earth and its rate of change with respect to time, the so-called secular variation (SV). The launch of low orbiting satellite missions (Ørsted, CHAMP, SWARM) dedicated to magnetic field measurements indeed presented a huge leap on the quality and coverage of measured data (see for instance Finlay et al., 2017).





In this context, the development of geomagnetic data assimilation (DA) algorithms is timely. DA consists in the estimation of a model state trajectory using (i) a numerical model that advects the state in time and (ii) measurements used to correct its trajectory. DA algorithms can be split in two main families: sequential methods that alternate between forecast (integration of the forward model) and analysis (statistical state inference from observations) steps, and variational methods that minimise

the misfit between the observations and model state predictions over the whole considered time-span by tuning the initial state conditions and model parameters by means of adjoint equations. Both types of algorithms are already commonplace in meteorology and oceanography, but have only been recently introduced in geomagnetism (for details, see Fournier et al., 2010; Gillet, 2019).

In this article, we present a Python package called `pygeodyn` devoted to geomagnetic DA based on a sequential method,
namely an augmented state Kalman Filter (see Evensen, 2003). It uses a reduced numerical model of the core surface dynamics that allows to alleviate the computation time inherent to DA algorithms. The reduced model is based on stochastic Auto-Regressive processes of order 1 (AR-1 processes). These are anchored to cross-covariances derived from three-dimensional numerical geodynamo simulations. We provide examples involving the 'coupled-earth' (Aubert et al., 2013) and 'mid-path' (Aubert et al., 2017) dynamos.

The aim of `pygeodyn` is to provide the community with a tool that can easily be used or built upon. It is made to ease the updating of data and the integration of new numerical models, for instance to test them against geophysical data. This way, it can be compared with other existing DA algorithms (e.g. Baerenzung et al., 2018).

The manuscript is organised as follows: Section §2 presents the principles under which the `pygeodyn` package was developed. Section §3 is a technical description of the version 1.0.0 of the package (Huder et al., 2019a) that also gives the basic
necessary scientific background (for details, see Barrois et al., 2017, 2018; Gillet et al., 2019). In §4 we give examples of the visualization interface `webgeodyn` to which is coupled the core surface DA tool `pygeodyn`. We discuss in §5 possible future developments and applications of this tool.

## 2    `pygeodyn` outlook

### 2.1    Principles

In order to support the use of DA in geomagnetism, the package is designed with the following characteristics in mind:

**Easy-to-use:**

- It is written in Python3, now a widespread language in the scientific community thanks to the NumPy/Scipy suites.

- It is based on few and classical dependencies.

- A README file documents the installation procedure; it only requires Python with NumPy and a Fortran compiler
(other dependencies are installed during the setup).

**Flexible:**


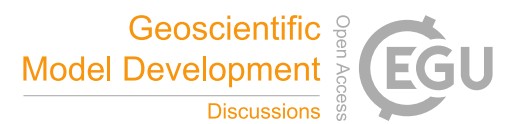

- Algorithm parameters can be tuned through configuration files and command line arguments.

- Algorithms are designed to be modular, in order to allow the independent use of their composing steps.

- Extension of the features is eased by readable open-source code (following PEP8) that is documented inline and online with Sphinx.

**Reproducible/stable:**

- The source code is versioned on a Git repository, with tracking of bugs and development versions with clear release notes.

- Unitary and functional tests are routinely launched by continuous integration pipelines. Most of the tests use the Hypothesis library[1] to cover a wide range of test cases.

- Logging of algorithm operations is done by default with the `logging` library.

**Efficient:**

- Direct integration of parallelisation is possible using Message Passing Interface (MPI)

- Lengthy computations (such as Legendre polynomial evaluations) are performed by Fortran routines wrapped in Python.

**Easy to post-process:**

- Output files are generated in HDF5 binary file format that is highly versatile (close to NumPy syntax) and more time- and size-efficient.

- The output format is directly supported by the visualisation package `webgeodyn` for efficient exploration of the computed data (see Section §4).

## 2.2 User profiles

The package was designed for several user types:

**Standard user:** the user will use the supplied DA algorithms with the supplied data. The algorithms can be modified through the configuration files so this requires almost no programming skill.

**Advanced user:** the user will use the supplied DA algorithms but wants to run it on its own data. In this case, the user needs to follow the documentation to implement the reading methods of the data. This requires a few Python programming skills and a basic knowledge of object-type structures.

**Developer user:** the user wants to design its own algorithm using the low-level functions implemented in the package. The how-to is also documented but it requires some experience in Python programming and object-type structures.

---

[1]https://hypothesis.works/





The documentation was written with these categories of users in mind. The README explains how to install and launch computations with the facilities provided for all kinds of users. For more advanced uses, an in-depth guide was written composed of two parts: the first one explains how to set up new data types as input (advanced users) whereas the second part is dedicated to developer users who want to use low-level features. This last part is also intended to be used with the developer documentation
that is generated with Sphinx and available online[2].

## 3   `pygeodyn 1.0.0` content

### 3.1   Model state

DA algorithms are to be supplied in the forms of subpackages for `pygeodyn`. The intention is to have interchangable algorithms and be able to easily expand the existing algorithms. In the version described in this article (0.8.0), we provide a
subpackage `augkf` that implements algorithms based on an augmented state Kalman filter (AugKF) initiated by Barrois et al. (2017). The algorithm takes for basis the induction equation at the core surface that we write in the spherical harmonic spectral domain as:

$$\dot{\boldsymbol{b}} = \mathbf{A}(\boldsymbol{b})\boldsymbol{u} + \boldsymbol{e}_r \,. \tag{1}$$

Vector $\boldsymbol{b}$ (resp. $\boldsymbol{u}$ and $\dot{\boldsymbol{b}}$) stores the (Schmidt semi-normalized) spherical harmonic coefficients of the magnetic field (resp. the
core flow and the SV) up to a truncation degree $L_b$ (resp. $L_u$ and $L_{sv}$). The number of stored coefficients in those vectors are respectively $N_b = L_b(L_b + 2)$, $N_u = 2L_u(L_u + 2)$ and $N_{sv} = L_{sv}(L_{sv} + 2)$. $\mathbf{A}(\boldsymbol{b})$ is the matrix of Gaunt-Elsasser integrals (Moon, 1979) of dimensions $N_{sv} \times N_u$, depending on $\boldsymbol{b}$. The vector $\boldsymbol{e}_r$ stands for errors of representativeness (of dimension $N_{sv}$). This term accounts for both subgrid induction (arising due to the truncation of the fields) and magnetic diffusion. Quantities $\boldsymbol{b}(t)$, $\boldsymbol{u}(t)$ and $\boldsymbol{e}_r(t)$ describe the model state $\boldsymbol{X}(t)$ at a given epoch $t$ on which algorithm steps act.
The model states are stored as a subclass of NumPy array called `CoreState` (implemented in `corestate.py`). The subclass allows efficient storing and easy access to the coefficients for $\boldsymbol{b}$, $\boldsymbol{u}$, $\boldsymbol{er}$ and $\dot{\boldsymbol{b}}$ but can also include additional quantities if needed.

### 3.2   Algorithm steps

The sequential DA algorithm is composed of two kinds of operations:

**Forecasts** are performed every $\Delta t_f$. An ensemble of $N_e$ core states is time stepped between $t$ and $t + \Delta t_f$.

    **Analyses** are performed every $\Delta t_a$ with $\Delta t_a = n\Delta t_f$ (analyses are performed every $n$ forecasts). The ensemble of core states at $t_a$ is adjusted by performing a Best Linear Unbiased Estimate (BLUE) using observations at $t = t_a$.

---

[2]https://geodynamo.gricad-pages.univ-grenoble-alpes.fr/pygeodyn/index.html



These steps require spatial cross-covariances that are derived from geodynamo runs (referred to as priors, see §3.3.3). Realizations associated with those priors are noted $\boldsymbol{b}^*$, $\boldsymbol{u}^*$ and $\boldsymbol{e}_r^*$ for respectively the magnetic field, the core flow and errors of representativeness.

From a technical point of view, algorithm steps take `CoreState` objects as inputs and return the `CoreState` resulting
from the operations. Forecasts and analyses are handled by the `Forecaster` and `Analyser` modules that are implemented in the `augkf` subpackage according to the AugKF algorithm.

### 3.2.1 Forecast and AR(1) processes

The forecast step consists in time-stepping $\boldsymbol{X}(t)$ between two epochs. AR-1 processes built on geodynamo cross-covariances are used to forecast $\boldsymbol{u}(t)$ and $\boldsymbol{e}_r(t)$. We write $\boldsymbol{u}(t) = \boldsymbol{u}_0 + \boldsymbol{u}'(t)$, with $\boldsymbol{u}_0$ the background flow (temporal average from the
geodynamo run) – and similar notations for $\boldsymbol{e}_r(t)$. Their numerical integration is based on an Euler-Maruyama scheme, which takes the form

$$
\left\{
\begin{array}{ll}
\boldsymbol{u}'(t + \Delta t_f) & = \boldsymbol{u}'(t) - \Delta t_f \mathbf{D}_u \boldsymbol{u}'(t) + \sqrt{\Delta t_f} \boldsymbol{\zeta_u}(t) \\
\boldsymbol{e}_r'(t + \Delta t_f) & = \boldsymbol{e}_r'(t) - \Delta t_f \mathbf{D}_e \boldsymbol{e}_r'(t) + \sqrt{\Delta t_f} \boldsymbol{\zeta_e}(t)
\end{array}
\right. .
\tag{2}
$$

$\mathbf{D}_u$ is the drift matrix for $\boldsymbol{u}$. $\boldsymbol{\zeta_u}$ is a Gaussian noise, uncorrelated in time and constructed such that spatial cross-covariances $\mathbf{P}_{uu} = \mathbb{E}\left(\boldsymbol{u}'\boldsymbol{u}'^T\right)$ of $\boldsymbol{u}$ match those of the prior geodynamo samples $\boldsymbol{u}^*$. $\mathbb{E}(\dots)$ stands for statistical expectation. Similar ex-
pressions and notations holds for $\boldsymbol{e}_r$. Note that $\boldsymbol{u}$ and $\boldsymbol{e}_r$ are supposed independent, which is verified for numerical simulations.

Drift matrices are estimated with different manners depending on the characteristics of the considered geodynamo priors. In the case where the geodynamo series do not allow to derive meaningful temporal statistics (e.g. too few samples, or simulations parameters leading to relatively too slow Alfvén waves, see Schaeffer et al., 2017), the two drift matrices are simply diagonal, and controlled by a single free parameter ($\tau_u$ for $\boldsymbol{u}$ and $\tau_e$ for $\boldsymbol{e}_r$):

$$\mathbf{D}_u = \frac{1}{\tau_u}\mathbf{I}_u \text{ and } \mathbf{D}_e = \frac{1}{\tau_e}\mathbf{I}_e, \tag{3}$$

with $\mathbf{I}_u$ (resp. $\mathbf{I}_e$) the identity matrix of rank $N_u$ (resp. $N_e$) The drift matrices being diagonal, the process is hereafter referred to as 'diagonal' AR-1. Barrois et al. (2017, 2019) used such diagonal AR-1 processes, based on the 'coupled-earth' dynamo simulation.

In the case where geophysically meaningful temporal statistics can be extracted from geodynamo samples, time cross-
covariance matrices

$$
\left\{
\begin{array}{ll}
\mathbf{P}_{uu^+} & = \mathbb{E}\left(\boldsymbol{u}'(t)\boldsymbol{u}'(t + \Delta t^*)^T\right) \\
\mathbf{P}_{ee^+} & = \mathbb{E}\left(\boldsymbol{e}_r'(t)\boldsymbol{e}_r'(t + \Delta t^*)^T\right)
\end{array}
\right. ,
\tag{4}
$$

are derived according to a sampling time $\Delta t^*$. $\mathbf{D}_{u,e}$ are then defined as (see Gillet et al., 2019, for details and an application to the 'mid-path' dynamo):

$$
\left\{
\begin{array}{ll}
\mathbf{D}_u & = \dfrac{\mathbf{I}_u - (\mathbf{P}_{uu}^{-1}\mathbf{P}_{uu^+})^T}{\Delta t^*} \\
\mathbf{D}_e & = \dfrac{\mathbf{I}_e - (\mathbf{P}_{ee}^{-1}\mathbf{P}_{ee^+})^T}{\Delta t^*}
\end{array}
\right. .
\tag{5}
$$





$\mathbf{D}_{u,e}$ are now dense, hence processes using this expression are referred to as 'dense' AR-1 processes.

The first step of the forecast is to compute $\boldsymbol{u}(t+\Delta t_f)$ and $\boldsymbol{e_r}(t+\Delta t_f)$ using Eqs. (2) (with matrices depending on the AR-1 process type). Then, the vector $\boldsymbol{b}(t+\Delta t_f)$ is evaluated thanks to Eq. (1) by using $\boldsymbol{u}(t+\Delta t_f)$, $\boldsymbol{e_r}(t+\Delta t_f)$ and $\boldsymbol{b}(t)$ with an explicit Euler scheme:

$$\boldsymbol{b}(t+\Delta t_f) = \boldsymbol{b}(t) + \Delta t_f \left[\mathbf{A}(\boldsymbol{b}(t))\boldsymbol{u}(t+\Delta t_f) + \boldsymbol{e_r}(t+\Delta t_f)\right].\tag{6}$$

This yields the forecast core state $\boldsymbol{X}^f(t+\Delta t_f)$. As this step is performed independently for every realization, realizations can be forecast in parallel. This is implemented in supplied algorithms with a MPI scheme.

### 3.2.2 Analysis

The analysis step takes as input the ensemble of forecast core states $\boldsymbol{X}^f(t_a)$, the geodynamo statistics, plus main field and SV
observations at $t = t_a$ together with their uncertainties. It is performed in two steps:

(i) First, a BLUE of an ensemble of realisations of $\boldsymbol{b}$ is performed from magnetic field observations $\boldsymbol{b}^o(t)$ and the ensemble of forecasts $\boldsymbol{b}^f(t)$ using the forecast cross-covariance matrix for $\boldsymbol{b}$.

(ii) Second, a BLUE of an ensemble of realisations of the augmented state $\boldsymbol{z} = [\boldsymbol{u}^T \boldsymbol{e_r}^T]^T$ is performed from SV observations $\dot{\boldsymbol{b}}^o(t)$, the ensemble of analysed main field from step (i), and the ensemble of forecasts for $\boldsymbol{u}^f(t)$ and $\boldsymbol{e}_r^f(t)$, using a
forecast cross-covariance matrix for $\boldsymbol{z}$.

For more details on the above steps, we refer to Barrois et al. (2017, 2018, 2019); Gillet et al. (2019).

### 3.3 Input data

#### 3.3.1 Command line arguments

Computations can be launched by running `run_algo.py` that accepts several command line arguments. These arguments
and their default value (taken if not supplied) are given in Table 1. The first group corresponds to the computation parameters, the only non-optional parameter being the path to the configuration file. The second group of parameters is linked to the output files: name of data files and logs. We stress the importance of the argument −m that fixes the ensemble size $N_e$, meaning the number of realisations on which the Kalman filter will be performed. As $N_e$ forecasts are performed at each epoch, this value has an important impact on the computation time (see §3.4). It is advised to set it to at least 20 to get a converged measure of
the dispersion within the ensemble of realizations.

#### 3.3.2 Configuration file

The `pygeodyn` configuration file allows to set the values of quantities used in the algorithm called *parameters*. This configuration file is a text-file containing three columns: one for the parameter name, one for the type and one for the parameter value. We refer to Table 2 for the list of parameters that can be set this way.



**Table 1.** Command line arguments of `pygeodyn/run_algo.py`

| Argument | Default value | Description |
|---|---|---|
| -conf | none | Path to the configuration file (see Sec. 3.3.2) |
| -algo | augkf | Name of the algorithm to use |
| -m | 10 | Ensemble size (number of realisations) to consider in the computation |
| -seed | Random | Seed used to initialize NumPy random state |
| -path | User_directory/pygeodyn_results | Path where the folder containing the output files will be created |
| -cname | Current_computation | Name of the folder that will be created to store the output files |
| -l | none | Name of the logfile |
| -v | 2 | Verbosity level of the log (1: debug, 2: info, 3: warning, 4: error, 5: critical) |

The first group of parameters sets the number of coefficients to consider for the core state quantities and the Legendre polynomials that are used to evaluate the Gaunt-Elsasser integrals that enter $\mathbf{A}(\boldsymbol{b})$.

The second group sets the time-span: starting time $t_{start}$, final time $t_{end}$, and time intervals (in months) for forecasts $\Delta t_f$ and analyses $\Delta t_a$ that were already addressed. To avoid imprecise decimal representation, the times are handled with NumPy's

`datetime64` and `timedelta64` classes (e.g. January 1980 is '1980-01'[3]).

Parameters of the third group tune the AR-1 processes used in the forecasts. `ar_type` can be set to `diag` (in this case, $\tau_u$ and $\tau_e$ will be used as in Eq. (3)) or to `dense` (in this case, $\Delta t^\star$ will be used to sample the prior data and compute drift matrices with Eqs. (5)).

The fourth group allows to trigger the use of a principal component analysis (PCA) for the core flow. By setting $N_{pca}$, the

algorithm will perform forecasts and analyses on the subset composed of the first $N_{pca}$ principal components describing the core flow (stored by decreasing explained variance), rather than on the entire core flow model. This is advised when using dense AR-1 processes (see Gillet et al., 2019). The normalisation of the PCA can be modified by setting `pca_norm` to `energy` (so that the variance of each principal component is homogeneous to a core surface kinetic energy) or to `None` (PCA performed directly on the Schmidt semi-normalized core flow Gauss coefficients).

The fifth group allows to change the initial conditions of the algorithm. By setting `core_state_init` to `constant`, all realisations of the initial core state will be equal to the average prior. If set to `normal`, realisations of the initial core state will be drawn according to a normal distribution centered on the dynamo prior average, within the dynamo prior cross-covariances (default behaviour). It is possible to set the initial core state to the core state from a previous computation by setting `core_state_init` to `from_file`. In this latter case, the full path of the hdf5 file of the previous computation and the

date of the core state to use must be given (`init_file` and `init_date`).

The last group of parameters allow to set the types of input data (priors and observations) that are presented in more details in the next section.

---

[3]More precisely, January 1st 1980





**Table 2.** Parameters available in a `pygeodyn` configuration file.

| Parameter | Name in the file | Description |
|---|---|---|
| $L_b$ | Lb | Maximal spherical harmonic degree of the magnetic field |
| $L_u$ | Lu | Maximal spherical harmonic degree of the core flow |
| $L_{sv}$ | Lsv | Maximal spherical harmonic degree of the secular variation |
| $N_\theta$ | N_theta | Number of angles to use for the evaluation of Legendre polynomials |
| $t_{start}$ | t_start | Starting time for the algorithm |
| $t_{end}$ | t_end | Ending time for the algorithm |
| $\Delta t_f$ | dt_f | Time step between forecasts |
| $\Delta t_a$ | dt_a | Time step between analyses |
| AR type | ar_type | Type of Auto-Regressive (AR) process to use in forecasts (see Sec. 3.2.1) |
| $\tau_u$ | TauU | Time constant for the core flow diagonal AR-1 |
| $\tau_e$ | TauE | Time constant for the subgrid errors diagonal AR-1 |
| $\Delta t^\star$ | dt_sampling | Sampling timestep for computing dense AR(1) matrices |
| $N_{pca}$ | N_pca | Number of Principal Components (PC) for PCAnalysis of the core flow |
| PCA normalisation | pca_norm | Normalisation to use on the core flow on which the PCA is performed |
| Type of initialisation | core_state_init | Method to use to generate the initial core state |
| File to use for initialisation | init_file | Complete path of the file containing the desired initial core state |
| Date to use for initialisation | init_date | Date of the desired initial core state |
| Prior directory | prior_dir | Directory containing the prior data |
| Prior type | prior_type | Type of the prior data |
| Obs directory | obs_dir | Directory containing the observation data |
| Obs type | obs_type | Type of the observation data |

### 3.3.3 Priors

Priors are composed of a series of snapshot core states that allow to estimate the background states and the cross-covariance matrices. The mandatory priors are those for the magnetic field $b$, the core flow $u$ and the subgrid errors $e_r$ that allow to derive the respective cross-covariance matrices $\mathbf{P}_{uu}$, $\mathbf{P}_{ee}$, etc.

5    The aforementioned snapshots currently come from geodynamo simulations, meaning that the covariance matrices for $b$, $u$, and $e_r$ will reflect the characteristics of the simulations. As a consequence, the forecasts will be done according to the statistics of the dynamo simulations. As examples, `pygeodyn` comes with two prior types derived from two simulations:

– `coupled_earth` from Aubert et al. (2013)



– `midpath` from Aubert et al. (2017)

Technically, the two types are interchangeable. However, only the `midpath` prior type allows the use of dense AR-1 processes as it requires time-correlations that cannot be extracted from `coupled_earth` runs.

### 3.3.4 Observations

Observations are measurements of the magnetic field and of the SV at a set of dates. These observations are used in the analysis step to perform the BLUE of the core state (see §3.2.2). `pygeodyn` provides two types of observations:

– `covobs`: Gauss coefficients and associated uncertainties at a series of epochs (every 6 months from 1840 to 2015), from the COV-OBS.x1 model derived by Gillet et al. (2015).

– `go_vo`: Ground-based observatory (GO) and virtual observatory (VO) data ($B_r$, $B_\theta$, $B_\phi$) and their associated uncer-
tainties. VO gather in one location at satellite altitude observations recorded by the spacecrafts around this site. GO are provided every 4 months from March 1997 onward for ground-based series, and VO every 4 months from March 2000 onward for virtual observatories. The satellite data come from the CHAMP and SWARM missions. Both VO and GO are cleaned as much as possible from external sources (for details, see Barrois et al., 2018).

In the code, observation data are to be supplied with the observation operator and errors in the form of a `Observation` ob-
ject. This allows to have a consistent interface between spectral data (`covobs`) and data recorded in the direct space (`go_vo`).

### 3.3.5 Beyond the supplied data

For advanced users, `pygeodyn` allows to define custom prior and observations types by supplying new data reading methods in the dedicated `pygeodyn` modules. Defining a custom prior type allows to use custom geodynamo simulation data to compute covariance matrices that will be used in the forecasts and analyses steps. Similarly, a new observation type can be supplied
with custom observation data that will be used to estimate the core state in the analysis step.

In other words, an advanced user can completely control the input data of the algorithm to test new magnetic observations and/or new numerical models, and derive new predictions from them.

### 3.4 Runtime scaling

To reduce computation time, supplied algorithms use MPI to perform forecasts (§3.2.1) of different realizations in parallel.
Analysis steps are not implemented in parallel, as they require in one go the whole ensemble of realizations.

To assert the effect of this parallelisation, model computations were performed on a varying number of cores. The configuration for this benchmark re-analysis is the following: the AugKF algorithm with diagonal AR-1 processes, using $m = 50$ realizations over $t_{end} - t_{start} = 60$ years, $\Delta t_f = 6$ months and $\Delta t_a = 12$ months.

The results are displayed on Fig. 1 with runtimes varying between 1 and 8 hours. The power law fit appears to be close to
$1/n$ (with $n$ the number of cores), with an offset time of 889 seconds that is probably associated with the 60 analysis steps



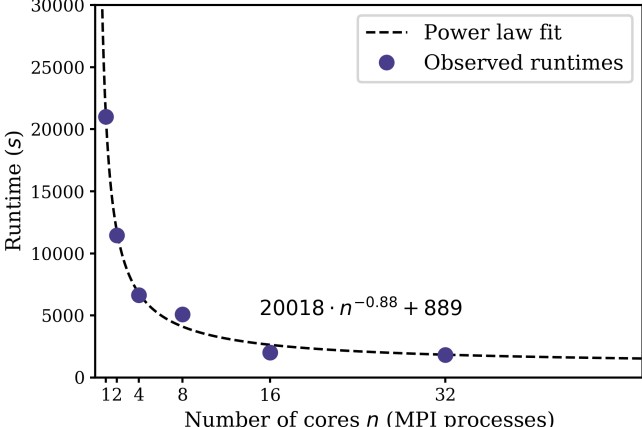

**Figure 1.** Evolution of runtime with respect to number of MPI processes (see text for details). Dots are the observed runtimes and the dashed line is a fit by the power law $t = a \cdot n^b + c$.

whose duration do not depend upon the number of cores. Note that the computations remain tractable whatever the number of cores. A basic sequential computation ($n = 1$) for a re-analysis using 50 realizations over 100 yrs is performed in less than half a day, while using 32 cores will reduce it to half an hour.

## 4 Visualisation

5  The format of `pygeodyn` output files is directly supported by the web-based visualisation package `webgeodyn` also developed in our group. The source code of this package is hosted at its own Git repository[4]. Being available on the Python package index, it can also be installed through the Python package installer `pip`.

   `webgeodyn` implements a web application with several modes of representation that allow to explore, display and diagnose the products of the re-analyses (Barrois et al., 2018). It is deployed at `http://geodyn.univ-grenoble-alpes.fr`

10  but can also be used locally on any `pygeodyn` data, once installed. We illustrate here several possibilities offered by the version 0.6.0 of this tool (Huder et al., 2019b).

### 4.1 Mapping on Earth's globe projections

Quantities at a given time can be displayed at the core surface in the *Earth's core surface* tab. Two representations can be used simultaneously: a streamdots/streamlines representation for the core flow components (orthoradial, azimuthal and norm) 15  and a color plot for all quantities (core flow horizontal divergence and components included). A timeslider allows to change the epoch at which the quantities are evaluated. Figure 2 shows an example with magnetic field as color plot and norm of the

---

[4]https://gricad-gitlab.univ-grenoble-alpes.fr/Geodynamo/webgeodyn



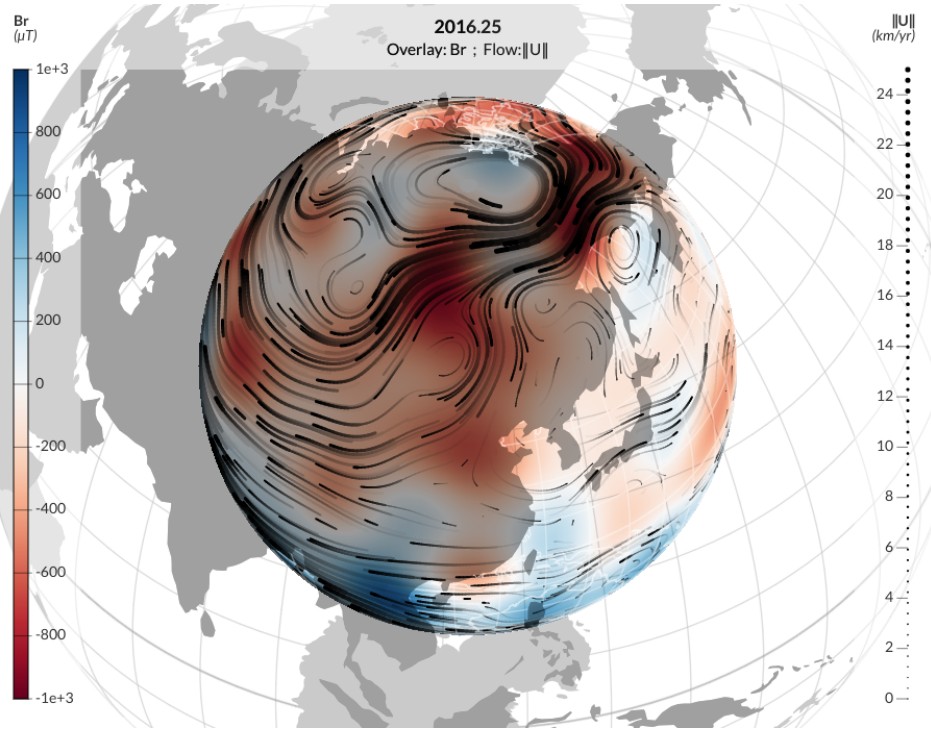

**Figure 2.** Example of map for the magnetic field and the flow at the core surface, obtained using `webgeodyn`, for the model calculated by Barrois et al. (2019). Here the radial magnetic field (colorscale) and streamlines for the core flow (black lines, which thickness indicates the intensity) are evaluated in 2016 from VO derived from the SWARM data.

core flow for the stream lines, for a re-analysis of VO and GO data using a diagonal AR-1 model. The plot is interactive with zooming, exporting (as pictures or animations) and display tuning features.

### 4.2 Time-series of harmonic coefficients

In the *Spherical Harmonics* tab, it is possible to look at the time evolution of a single spherical harmonic coefficient for a given
5    quantity (core flow, magnetic field, SV), or of the length-of-day. Several models can be displayed at once for comparison. Figure 3 shows the time evolution of one SV coefficient from a re-analysis of SV Gauss coefficient data using a dense AR-1 model. The interface gives the possibility to also represent the contribution from $e_r$. It is possible to zoom on the plot and export it as a picture or raw CSV data.

### 4.3 Comparison with ground-based and virtual observatory data

10    Computed data can be easily compared with the geomagnetic observations used for the analysis in the *Observatory data* tab. It allows to display the spatial components (radial, azimuthal and ortho-radial) of the magnetic field and its SV recorded by observatories. These can be either GO or VO data. Data can be displayed by clicking on a location on the globe and be





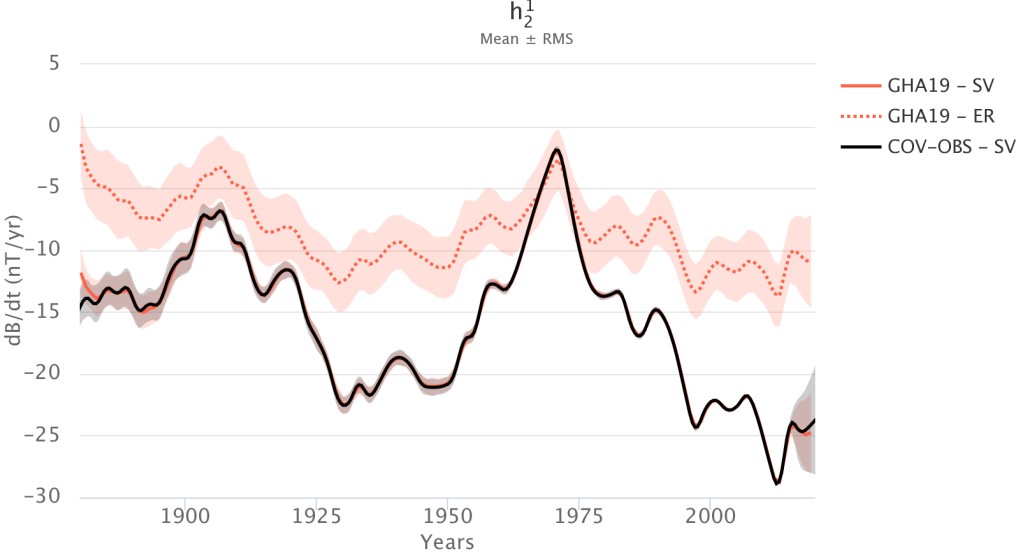

**Figure 3.** Time series for the SV spherical harmonic coefficient $h_2^1$ using `webgeodyn`. In red ('GHA19') the re-analysis by Gillet et al. (2019), obtained from the COV-OBS.x1 observations (in black) by Gillet et al. (2015). The solid lines represent the ensemble average, and the shaded areas the $\pm 1\sigma$ uncertainties. The dotted line gives the contribution from $\boldsymbol{e}_r$.

compared with spectral model data predictions evaluated at the observatory location. Figure 4 shows an example of the SV at a ground-based site in South America. One can compare how predictions from a re-analysis, together with its associated uncertainties, follow geophysical data (black dots) – here the model by Barrois et al. (2019), which uses a diagonal AR-1 model from GO and VO series. It can also be used to compare predictions from several magnetic field models – here COV-OBS.x1, which is constrained by magnetic data only up to 2014.5.

On top of the three examples illustrated above, the package `webgeodyn` also gives the possibility to display and export Lowes-Mauersberger spatial spectra, or cross-sections at the core surface as a function of time and longitude (respectively latitude) for a given latitude (respectively longitude).

## 5 Conclusions

We presented the Python toolkit `pygeodyn` that allows to:

– calculate models of the flow at the core surface from SV Gauss coefficient data

– calculate models of the flow and the magnetic field at the core surface from measurements of the magnetic field and its SV above the Earth's surface

– represent and analyse the results via the web interface `webgeodyn`.



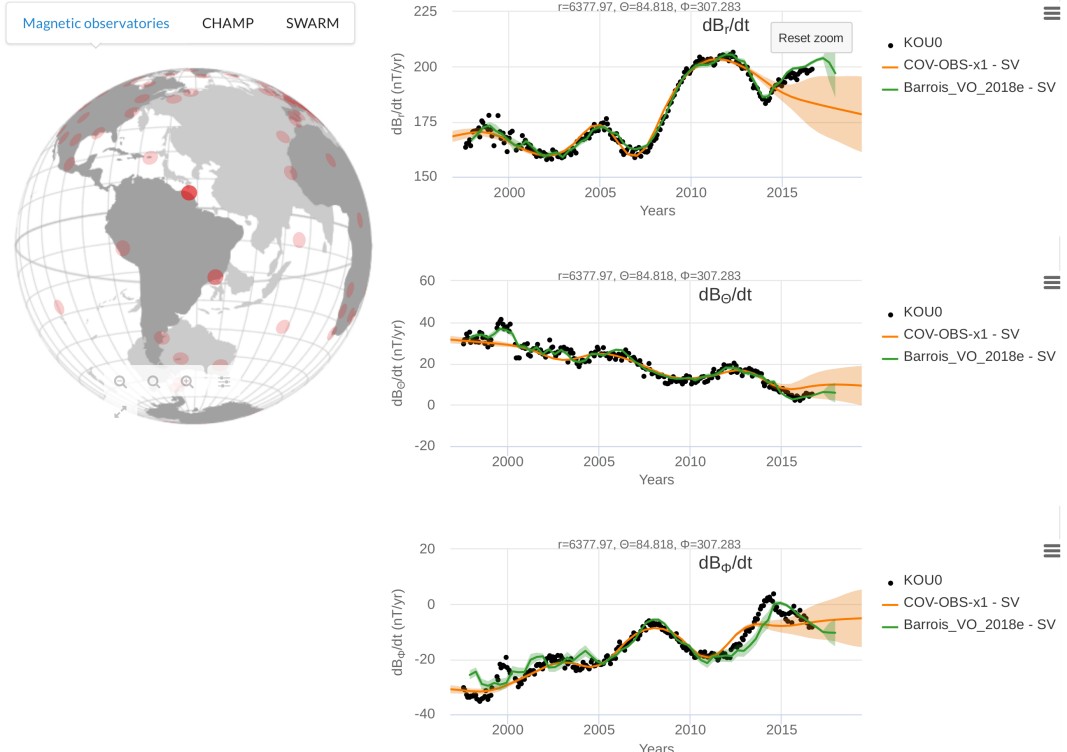

**Figure 4.** Time series of the three components of the SV (in spherical coordinates) at the Kourou observatory (French Guyana, location in dark red on the globe on the left) using `webgeodyn`. The green line ('Barrois_VO_2018e') is from the core surface re-analysis by Barrois et al. (2019), using as observations GO and VO data, and a diagonal AR-1 model. For comparison are also shown (in orange) the COV-OBS.x1 model predictions at this site. The solid lines are the ensemble mean, and the shaded areas represent the $\pm 1\sigma$ uncertainties.

The underlying algorithm relies on AR-1 stochastic processes to advect the model in time. It is anchored to statistics (in space and optionally in time) from free runs of geodynamo models. It furthermore accounts for errors of representativeness due to the finite resolution of the magnetic and velocity fields.

This Python tool has been designed with several purposes in mind, among which :

– test of the Earth-likeness of geodynamo models

– comparison with alternative geomagnetic DA algorithms

– production of magnetic models under some constraints from the core dynamics

– education of students on issues linked to core dynamics and geomagnetic inverse problem.



*Code and data availability.* The version 1.0.0 of `pygeodyn` is archived on Zenodo (Huder et al., 2019a). Other versions are available on the Git repository located at https://gricad-gitlab.univ-grenoble-alpes.fr/Geodynamo/pygeodyn. The version 0.6.0 of `webgeodyn` used for the plots is archived at https://doi.org/10.5281/zenodo.2645025. This version (and others) can be installed from the Git repository (https://gricad-gitlab.univ-grenoble-alpes.fr/Geodynamo/webgeodyn) or as a Python package with `pip`. Data sets can be downloaded on http://geodyn.univ-grenoble-alpes.fr

*Author contributions.* The scientific development of the presented package was done by NG and LH. The technical development (including tests and packaging) was done by LH and FT. All authors contributed to the writing of the manuscript.

*Competing interests.* The authors declare no competing interests.

*Acknowledgements.* We thank Julien Aubert for supplying the geodynamo series used as priors, and Christopher Finlay for providing the `go_vo` data. We are grateful to Nathanaël Schaeffer for helpful comments on the manuscript. This work is partially supported by the French Centre National d'Etudes Spatiales (CNES).



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
