# Peer review of "pygeodyn 1.1.0: a Python package for geomagnetic data assimilation"

_Geoscientific Model Development, 2019_

## Referee Comment (RC1) · Ciaran Beggan (Referee) · 8 May 2019

This manuscript by Huder, Gillet and Tholland explains the Python code for accessing and executing the code for their geodynamo data assimilation code. The scientific work based on this code has been published in previous papers (Barrois et al, 2017/2018, for example) and so it is to be applauded that they have made the great effort to allow others to reproduce their scientific research in an open manner.

The visualisation tools on the webgeodyn site are also very impressive, easy to use and simple to understand and almost deserve a paper on their own merit. I spent a long time examining the different plots and combinations of data that can be placed together. Figure 2 is particularly stunning (on the website, it can be rotated and animated).

[Printer-friendly version]{.underline}

[Discussion paper]{.underline}

[Figure]

The manuscipt is probably of most use to researchers interested in the details of how to recreate data assimilation within the scope of the geodynamo snapshots though the authors have made it clear that a user can access the code at the different levels of expertise required. In reality, it is a companion paper to the Barrois et al and Aubert papers of the past five years. The parameters and assumptions are well explained and the flow of the work load is clear.

I have no major comments or suggestions to make. I would however, suggest that the authors revise their frequent use of the phrase 'allows to' which appears several times. For example on page 7, line 15, they could write: ' The fifth group allows the user to ...' or on page 8, line 1: '... states that allow an estimate of the background states ...'.

Minor corrections:

Page 1, line 9: generated by motion of the liquid

Page 1, line 14: that DNS are not capable of reproducing changes

Page 3, line 24: but wants to run their own data. (and again on Line 27)

Page 4, line 11: takes as input? (rather than basis)

Page 5, line 8: consists of time-stepping

Table 1: why is the default -m parameter equal to 10 in the Table but in the text you recommend 20?

Page 7, line 4: I'm slightly confused about the decimal representation explanation - surely the input magnetic or SV data are monthly means (or 4 monthly VO) so it hardly matters how the date is represented at 64-bit precision - we're not at the microsecond level of precision. It's OK if that is the default Python class to use.

Figure 1: Is it better to have runtime on a log scale to emphasis the point?

Page 10, line 5: output files are directly

Acknowledgement: Perhaps add a note about the use of INTERMAGNET data (which I assume are used for the plots?)

---

## Referee Comment (RC2) · Anonymous Referee #2 · 22 May 2019

**Review of Manuscript # gmd-2019-82**
pygeodyn 1.0.0: a Python package for geomagnetic data assimilation

Recommendation: accepted subject to minor revisions

The manuscript presents a python software package/library for the simulation and data assimilation of geomagnetic models. The packages provides a surface dynamic model, a reduced order model based on autoregressive processes, geomagnetic observations, and an data assimilation method (the augmented Kalman filter) in a single package. All of the results in the paper are easily reproduced by downloading pygeodyn and the plotting package webgeodyn also developed by the same group. Although there are some significant deficiencies in the software package itself, as well as a lack of accessible user manual, the paper present a comprehensive description of the default features and data. I recommend accepting the paper after some minor revisions are done. The software on the other hand, is far from ready for widespread user adoption. If the main purpose is to make this package accessible, I strongly urge the developers to provide a human-readable user manual, tutorial, and customization examples.

Major Comments:

1. My major concern is that the software is far from ready for use by non-advanced python expert. The code is written is such a way that it's all but impossible to read and understand, much less modify to include new data, models, assimilation techniques. Worst of all, there is no proper documentation detailing the structure of the package, objects being used, and organization of the assimilation system. These are indispensable elements for customization and none are present. If the developers really want a widespread adoption of pygeodyn, then they need to work hard on making the software accessible and well documented. To be completely sincere, I wouldn't recommend this package to anyone in the geosciences community.

2. The git repository should only be used for the python software and not for the data. I strongly suggest that the data be stored on a separate repository or server since it is over one gigabyte of data. It makes no sense to store the data together with the python scripts.

Minor Comments:

1. On the Introduction (page 2, lines 3–6) the authors mention that there are two main families: sequential and variational. This might be an oversimplification
since the 3D-Var is a variational method that is sequential, and ensemble Kalman smoother is not a variational that is a smoother. I suggest the authors rework this sentence since it's misleading.

2. In section 2.2, the authors further classify the type of users for pygeodyn. As stated above, the software is far from ready for customization so I would suggest the authors rework or remove this section since it would be misleading to claim that the python package is accessible, it is not.

3. The proper websites of where to download pygeodyn is buried at the very end of the paper. I strongly suggest this be moved in the forefront, maybe at the end of the introduction.

4. increase the font size on the axis and labels for Figure 4

---

## Author Comment (AC1) · 5 Jul 2019

This manuscript by Huder, Gillet and Tholland explains the Python code for accessing and executing the code for their geodynamo data assimilation code. The scientific work based on this code has been published in previous papers (Barrois et al, 2017/2018, for example) and so it is to be applauded that they have made the great effort to allow others to reproduce their scientific research in an open manner.

The visualisation tools on the webgeodyn site are also very impressive, easy to use and simple to understand and almost deserve a paper on their own merit. I spent a long time examining the different plots and combinations of data that can be placed together. Figure 2 is particularly stunning (on the website, it can be rotated and animated).

[Figure]

The manuscipt is probably of most use to researchers interested in the details of how to recreate data assimilation within the scope of the geodynamo snapshots though the authors have made it clear that a user can access the code at the different levels of expertise required. In reality, it is a companion paper to the Barrois et al and Aubert papers of the past five years. The parameters and assumptions are well explained and the flow of the work load is clear.

⇒ **We thank the referee for his positive remarks.**

I have no major comments or suggestions to make. I would however, suggest that the authors revise their frequent use of the phrase 'allows to' which appears several times. For example on page 7, line 15, they could write: ' The fifth group allows the user to ...'or on page 8, line 1: '... states that allow an estimate of the background states ...'.

⇒ **Following the referee's suggestion, the wording was changed in the mentioned sentences and in a few others.**

Minor corrections:

- Page 1, line 9: generated by motion of the liquid

  ⇒ **Corrected.**

- Page 1, line 14: that DNS are not capable of reproducing changes

  ⇒ **Corrected.**

- Page 3, line 24: but wants to run their own data. (and again on Line 27)

  ⇒ **Corrected.**

- Page 4, line 11: takes as input? (rather than basis)

> ⇒ **Reworded as 'The algorithm is based on the radial induction equation...'**

- Page 5, line 8: consists of time-stepping

  > ⇒ **Corrected.**

- Table 1: why is the default -m parameter equal to 10 in the Table but in the text you recommend 20?

  > ⇒ **This is an outdated value that was used for tests. We thank the referee for pointing that out. The default parameter is set to 20 in the new version (1.1.0).**

- Page 7, line 4: I'm slightly confused about the decimal representation explanation -surely the input magnetic or SV data are monthly means (or 4 monthly VO) so it hardlymatters how the date is represented at 64-bit precision - we're not at the microsecondlevel of precision. It's OK if that is the default Python class to use.

  > ⇒ **The 64-bit precision is indeed the default NumPy way to store dates. The point of using this class was not firstly to enhance precision but rather to use a dedicated tool to store dates that leaves no ambiguity. Our previous decimal representation could lead to questionning on days and month shifts (e.g.: is 1980.16667, the 1st, the 31st of January 1980, or the 1st of Feburary 1980?)**

- Figure 1: Is it better to have runtime on a log scale to emphasis the point?

  > ⇒ **Unfortunately, as there is a part of the runtime that does not depend on the number of cores, a log plot does not lead to a straight line and is not clearer in our opinion. Still, the units of the runtime (and of the fit) were changed from seconds to hours for readability. The text describing the graph was changed accordingly.**

- Page 10, line 5: output files are directly

    ⇒ **Corrected.**

- Acknowledgement: Perhaps add a note about the use of INTERMAGNET data (which I assume are used for the plots?)

    ⇒ **That is indeed the case. The acknowledgement was added.**

---

## Author Response (AR1)

Dear Editor,

Please find attached our revised version of the manuscript gmd-2019-82 entitled "pygeodyn 1.1.0: a Python package for geomagnetic data assimilation", by Huder, Gillet & Thollard, together with our response (in bold) to the points made by the referees (recalled in normal font).

The main modification in comparison with the manuscript initially submitted concerns the documentation of the package. Following referee 2's major comment, we indeed have collected and extended the documentation and uploaded a navigable version on the web. The version of pygeodyn was then upgraded to 1.1.0 that is the version described in the paper. The title of the article and Zenodo references have therefore been updated.

Do not hesitate to contact us if you need further information,

On behalf of the authors, Loïc Huder

**Response to Referee 1**

This manuscript by Huder, Gillet and Tholland explains the Python code for accessing and executing the code for their geodynamo data assimilation code. The scientific work based on this code has been published in previous papers (Barrois et al, 2017/2018, for example) and so it is to be applauded that they have made the great effort to allow others to reproduce their scientific research in an open manner.

The visualisation tools on the webgeodyn site are also very impressive, easy to use and simple to understand and almost deserve a paper on their own merit. I spent a long time examining the different plots and combinations of data that can be placed together. Figure 2 is particularly stunning (on the website, it can be rotated and animated).

The manuscipt is probably of most use to researchers interested in the details of how to recreate data assimilation within the scope of the geodynamo snapshots though the authors have made it clear that a user can access the code at the different levels of expertise required. In reality, it is a companion paper to the Barrois et al and Aubert papers of the past five years. The parameters and assumptions are well explained and the flow of the work load is clear.

**$\Rightarrow$ We thank the referee for his positive remarks.**

I have no major comments or suggestions to make. I would however, suggest that the authors revise their frequent use of the phrase allows to which appears several times. For example on page 7, line 15, they could write: The fifth group allows the user to ...or on page 8, line 1: ... states that allow an estimate of the background states ....

 $\Rightarrow$  Following the referee's suggestion, the wording was changed in the mentioned sentences and in a few others.

Minor corrections:

• Page 1, line 9: generated by motion of the liquid

 $\Rightarrow$  Corrected.

• Page 1, line 14: that DNS are not capable of reproducing changes

 $\Rightarrow$  Corrected.

• Page 3, line 24: but wants to run their own data. (and again on Line 27)

 $\Rightarrow$  Corrected.

- Page 4, line 11: takes as input? (rather than basis)
  - $\Rightarrow$  Reworded as 'The algorithm is based on the radial induction equation...'
- Page 5, line 8: consists of time-stepping

 $\Rightarrow$  Corrected.

- Table 1: why is the default -m parameter equal to 10 in the Table but in the text you recommend 20?
  - $\Rightarrow$  This is an outdated value that was used for tests. We thank the referee for pointing that out. The default parameter is set to 20 in the new version (1.1.0).
- Page 7, line 4: Im slightly confused about the decimal representation explanation -surely the input magnetic or SV data are monthly means (or 4 monthly VO) so it hardlymatters how the date is represented at 64-bit precision were not at the microsecondlevel of precision. Its OK if that is the default Python class to use.
  - ⇒ The 64-bit precision is indeed the default NumPy way to store dates. The point of using this class was not firstly to enhance precision but rather to use a dedicated tool to store dates that leaves no ambiguity. Our previous decimal representation could lead to questionning on days and month shifts (e.g.: is 1980.16667, the 1st, the 31st of January 1980, or the 1st of Feburary 1980?)
- Figure 1: Is it better to have runtime on a log scale to emphasis the point?

- $\Rightarrow$  Unfortunately, as there is a part of the runtime that does not depend on the number of cores, a log plot does not lead to a straight line and is not clearer in our opinion. Still, the units of the runtime (and of the fit) were changed from seconds to hours for readability. The text describing the graph was changed accordingly.
- Page 10, line 5: output files are directly
  - $\Rightarrow$  Corrected.
- Acknowledgement: Perhaps add a note about the use of INTERMAGNET data (which I assume are used for the plots?)
  - $\Rightarrow$  That is indeed the case. The acknowledgement was added.

**Response to Referee 2**

The manuscript presents a python software package/library for the simulation and data assimilation of geomagnetic models. The packages provides a surface dynamic model, a reduced order model based on autoregressive processes, geomagnetic observations, and an data assimilation method (the augmented Kalman filter) in a single package. All of the results in the paper are easily reproduced by downloading pygeodyn and the plotting package webgeodyn also developed by the same group. Although there are some significant deficiencies in the software package itself, as well as a lack of accessible user manual, the paper present a comprehensive description of the default features and data. I recommend accepting the paper after some minor revisions are done. The software on the other hand, is far from ready for widespread user adoption. If the main purpose is to make this package accessible, I strongly urge the developers to provide a human-readable user manual, tutorial, and customization examples.

**Major Comments:**

1. My major concern is that the software is far from ready for use by nonadvanced python expert. The code is written is such a way that its all but impossible to read and understand, much less modify to include new data, models, assimilation techniques. Worst of all, there is no proper documentation detailing the structure of the package, objects being used, and organization of the assimilation system. These are indispensable elements for customization and none are present. If the developers really want a widespread adoption of pygeodyn, then they need to work hard on making the software accessible and well documented. To be completely sincere, I wouldnt recommend this package to anyone in the geosciences community.

- ⇒ In fact, such information existed under the form of a README in the root folder, an advanced guide in the doc folder and the online developer documentation. Still, we agree that is was not easy enough to find nor organised enough. Following the referee's comments, we have totally reworked our documentation in the new version (1.1.0) in order to present more clearly all the needed information (Now online at https://geodynamo.gricad-pages.univ-grenoble-alpes.fr/ pygeodyn/ and offline in the docs folder after HTML generation). Namely:
  - The package READMEs were broken down in several sections that were also expanded.
  - These sections are now navigable online and comprise:
    - \* Installation instructions
    - \* A brief scientific overview of the algorithm
    - \* An expanded description of the run\_algo script (including structure)
    - \* Tutorials on the definition of new types and on the reuse of forecast/analysis steps
    - \* The developer API that was originally online

We are grateful to the referee for triggering this documentation rework that should be a big step towards the accessibity of pygeodyn.

- 2. The git repository should only be used for the python software and not for the data. I strongly suggest that the data be stored on a separate repository or server since it is over one gigabyte of data. It makes no sense to store the data together with the python scripts.
  - $\Rightarrow$  We followed the referee's suggestion by separating the package sources from the data, each having now their own repository. We provide the user the commands to either fetch only the sources or the complete package.

**Minor Comments:**

- 1. On the Introduction (page 2, lines 36) the authors mention that there are two main families: sequential and variational. This might be an oversimplification since the 3D-Var is a variational method that is sequential, and ensemble Kalman smoother is not a variational that is a smoother. I suggest the authors rework this sentence since its misleading.
  - $\Rightarrow$  We agree with the referee, and now present the two main families of DA tools as being the variational (minimizing a cost function) and statistical (based on Bayes rule) avenues,

**with references to the books by Kalnay (2003) and Evensen (2009).**

- 2. In section 2.2, the authors further classify the type of users for pygeodyn. As stated above, the software is far from ready for customization so I would suggest the authors rework or remove this section since it would be misleading to claim that the python package is accessible, it is not.
  - $\Rightarrow$  The end of the section was rewritten to integrate the rework of the documentation. We hope that this will improve the accessibility of the package.
- 3. The proper websites of where to download pygeodyn is buried at the very end of the paper. I strongly suggest this be moved in the forefront, maybe at the end of the introduction.
  - $\Rightarrow$  Actually the recommendation of the referee goes against the journal guidelines asking to put the links for download in the *Code and data availability* section.
- 4. increase the font size on the axis and labels for Figure 4
  - $\Rightarrow$  This figure was generated directly from the webtool as a demonstration and is therefore not easily manipulated. We increased the size of the figure to accommodate this but the customization of these plots is part of future development of the webtool. Note however that the raw data can be exported to do a plot with other plotting softwares.

[revised manuscript text omitted]